# Clinicopathological Significance of EBV-Infected Gastric Carcinomas: A Meta-Analysis

**DOI:** 10.3390/medicina56070345

**Published:** 2020-07-13

**Authors:** Jung-Soo Pyo, Nae-Yu Kim, Dong-Wook Kang

**Affiliations:** 1Department of Pathology, Daejeon Eulji University Hospital, Eulji University School of Medicine, Daejeon 35233, Korea; jspyo@eulji.ac.kr; 2Department of Internal Medicine, Daejeon Eulji University Hospital, Eulji University School of Medicine, Daejeon 35233, Korea; naeyu46@eulji.ac.kr; 3Department of Pathology, Chungnam National University Sejong Hospital, 20 Bodeum 7-ro, Sejong 30099, Korea; 4Department of Pathology, Chungnam National University School of Medicine, 266 Munhwa Street, Daejeon 35015, Korea

**Keywords:** gastric carcinoma, Epstein–Barr virus, clinicopathological characteristics, histologic type, meta-analysis

## Abstract

*Background and objectives*: The present study aims to elucidate the clinicopathologic significance of Epstein–Barr virus (EBV) infection in gastric carcinomas (GCs) through a meta-analysis. *Materials and Methods*: Sixty-one eligible studies were included in the present meta-analysis. The included patients, with and without EBV infection, were 2063 and 17,684, respectively. We investigated the clinicopathologic characteristics and various biomarkers, including programmed death-ligand 1 (PD-L1) expression and tumor-infiltrating lymphocytes (TILs). *Results*: The estimated EBV-infected rate of GCs was 0.113 (95% confidence interval (CI): 0.088–0.143). The EBV infection rates in GC cells were 0.138 (95% CI: 0.096–0.194), 0.103 (95% CI: 0.077–0.137), 0.080 (95% CI: 0.061–0.106), and 0.042 (95% CI: 0.016–0.106) in the population of Asia, America, Europe, and Africa, respectively. There was a significant difference between EBV-infected and noninfected GCs in the male: female ratio, but not other clinicopathological characteristics. EBV infection rates were higher in GC with lymphoid stroma (0.573, 95% CI: 0.428–0.706) than other histologic types of GCs. There were significant differences in high AT-rich interactive domain-containing protein 1A (ARID1A) and PD-L1 expressions, and high CD8+ TILs between EBV-infected and noninfected GCs. *Conclusions*: Our results showed that EBV infection of GCs was frequently found in male patients and GCs with lymphoid stroma. EBV infection was significantly correlated with ARID1A and PD-L1 expressions and CD8+ TILs in GCs.

## 1. Introduction

The Epstein–Barr virus (EBV) is a ubiquitous human herpesvirus associated with several lymphoid and epithelial malignancies, including Burkitt’s lymphoma, Hodgkin’s lymphoma, nasal NK/T cell lymphoma, and a subset of gastric carcinomas (GCs) [1,2,3,4,5,6]. In 1990, Burke et al. first detected the EBV genomes in a small group of GCs using a polymerase chain reaction [1]. Shibata et al. demonstrated that EBV genomes were uniformly present in GC cells, resembling lymphoepithelioma cells [4]. After that, EBV involvement was detected not only in lymphoepithelioma-like GCs but also in a subset of ordinary GCs [4,7].

EBV-associated gastric cancers (EBVaGCs) have a unique molecular signature, which has defined this group of tumors as a distinctive molecular subtype of gastric cancer that accounts for approximately 10% of all GCs [2,3,4]. Thus, EBVaGC is the most common cancer among EBV-related malignancies. However, the prevalence of EBV infection in GCs has differed by reports and histologic subtypes [7,8,9,10,11,12,13,14,15,16,17,18,19,20,21,22,23,24,25,26,27,28,29,30,31,32,33,34,35,36,37,38,39,40,41,42,43,44,45,46,47,48,49,50,51,52,53,54,55,56,57,58,59,60,61,62,63,64,65,66,67]. Furthermore, cumulative information cannot be obtained from individual studies. As part of The Cancer Genome Atlas (TCGA) project, EBVaGCs are associated with distinct molecular changes, as follows: DNA hypermethylation, high frequency of *PIK3CA* mutation, *JAK2* gene amplification, programmed death-ligand 1/programmed cell death 1 ligand 2 (PD-L1/PD-L2) overexpression, and cyclin-dependent kinase inhibitor 2A (*CDKN2A*) silencing [2]. Recently, the loss of AT-rich interactive domain-containing protein 1A (ARID1A) was found in 20% of GCs and significantly correlated with EBVaGCs, PD-L1 status, as well as microsatellite instability (MSI) [64]. As the incidences and clinical features of GCs differ between regions, the clinicopathological characteristics of EBVaGCs may vary according to the various factors. In the present study, we investigate the clinicopathologic significance of EBVaGCs from eligible studies and perform the subgroup analysis to elucidate the EBV infection rate. We also evaluate the differences in the expression of various markers between EBVaGCs and non-EBVaGCs.

## 2. Materials and Methods

### 2.1. Published Study Search and Selection Criteria

Relevant articles were obtained by searching the PubMed database on 31 January 2020. For the search, the following keywords were used: “gastric carcinoma or gastric cancer or stomach cancer” and “Epstein–Barr virus or EBV”. The titles and abstracts of all searched articles were screened for the inclusion and exclusion of each article. Included articles contained information on the correlation between EBV positivity and clinicopathological characteristics in GCs. However, case reports, nonoriginal articles, or those not written in English were excluded from the present study. The PRISMA checklist is shown in Appendix A.

### 2.2. Data Extraction

Data associated with clinicopathological characteristics based on EBV positivity in GCs were extracted from each of the eligible studies [7,8,9,10,11,12,13,14,15,16,17,18,19,20,21,22,23,24,25,26,27,28,29,30,31,32,33,34,35,36,37,38,39,40,41,42,43,44,45,46,47,48,49,50,51,52,53,54,55,56,57,58,59,60,61,62,63,64,65,66,67]. Two independent authors obtained all the data. The data extracted were the author’s information, study location, number of patients analyzed, EBV-positive rates, and clinicopathological characteristics by EBV infection. Additional information on immunohistochemical stains is shown in Appendix A.

### 2.3. Statistical Analyses

The meta-analysis was performed using the Comprehensive Meta-Analysis software package 2.0 (Biostat, Englewood, NJ, USA). The EBV positivity rate was investigated in GCs. In addition, a subgroup analysis based on study location and histologic subtypes of GCs was performed. The correlations between EBV infection and clinicopathological characteristics were evaluated in GCs. In the present study, the following were included in the evaluated clinicopathological characteristics: age, sex, tumor size, tumor differentiation, histologic type, lymphatic, vascular, and perineural invasions, lymph node metastasis, and pTNM stages. Furthermore, the correlations between EBV positivity and p53, ARID1A, human epidermal growth factor receptor (HER2), and PD-L1 expressions, tumor-infiltrating lymphocytes (TILs), and microsatellite instability (MSI) in GCs were analyzed. We checked the heterogeneity between the studies by Q and *I*^2^ statistics, expressed as *p*-values. Additionally, we conducted a sensitivity analysis to assess the heterogeneity of the eligible studies and the impact of each study on the combined effects. In the meta-analysis, as the eligible studies used various populations, a random-effect model (rather than a fixed-effect model) was determined to be more suitable. The statistical difference between subgroups was evaluated by a metaregression test. We used Begg’s funnel plot and Egger’s test to assess the publication bias; if significant publication bias was found, the fail-safe N and trim-fill tests were additionally used to confirm the degree of publication bias. The results were considered statistically significant at *p* < 0.05.

## 3. Results

### 3.1. Selection and Characteristics of the Studies

In this study, 1301 relevant articles were found from the PubMed database and reviewed for a meta-analysis. Of these, 405 articles had no or a lack of sufficient information for the meta-analysis. A further 346 were excluded due to nonoriginal articles. Among the remaining articles, 489 reports were excluded for the following reasons: nonhuman studies (*n* = 238), articles on other diseases (*n* = 185), in a language other than English (*n* = 40), and duplication (*n* = 26); see Figure 1. Finally, 61 eligible articles were selected and included for the meta-analysis (Table 1). These studies included 19,747 GC patients with and without EBV infection (2063 and 17,684, respectively).

### 3.2. Epstein–Barr virus (EBV) Infected Rates of Gastric Carcinomas (GCs)

First, we investigated and analyzed the EBV-positive rates of GCs. The estimated EBV-positive rate was 0.113 (95% CI: 0.088–0.143) in overall GC cases. In the subgroup analysis based on study location, the EBV infected rate was the highest in Asia, compared to that in other regions. The EBV infected rate in the Asia region was 0.138 (95% CI: 0.096–0.194). In other areas, the EBV infected rates were 0.103, 0.080, and 0.042 in America, Europe, and Africa, respectively (Table 2).

### 3.3. Correlations Between Epstein–Barr virus (EBV) Infection and Clinicopathological Characteristics in Gastric Carcinomas (GCs)

The clinicopathological characteristics, according to EBV positivity, were investigated in GCs. The male patients showed a significantly higher estimation rate in the EBV-positive group than in the EBV-negative group (0.824 vs. 0.639; *p* < 0.001 in a metaregression test). Other clinicopathological characteristics, including age, tumor size, tumor differentiation, lymphatic, vascular, and perineural invasions, pT stage, lymph node metastasis, and pTNM stage, had no significant differences between EBV-infected and noninfected GCs (Table 3). Next, the EBV-positive rates by histologic type of GC were investigated (Table 4). The EBV-positive rate of GC with lymphoid stroma was 0.573 (95% CI: 0.428–0.706). This GC with lymphoid stroma showed higher EBV-positive rates compared to other tumor subtypes such as tubular adenocarcinoma (0.174), poorly cohesive carcinoma (0.078), papillary carcinoma (0.022), mucinous carcinoma (0.053), and undifferentiated carcinoma (0.111).

PD-L1 expressions in tumor and immune cells were significantly higher in EBVaGCs than in non-EBVaGCs (Table 5). In detail, PD-L1 expression rates of tumor cells were 0.573 (95% CI: 0.449–0.688) and 0.183 (95% CI: 0.118–0.272) in EBVaGCs and non-EBVaGCs, respectively. In addition, the PD-L1 expression rates of immune cells were 0.832 (95% CI: 0.630–0.935) and 0.487 (95% CI: 0.357–0.619) in EBVaGCs and non-EBVaGCs, respectively. ARID1A was highly expressed in EBVaGCs compared to non-EBVaGCs (0.29 vs. 0.170; *p* = 0.021 in a metaregression test). HER2 expression was higher in non-EBVaGCs than in EBVaGCs (0.104 vs. 0.048), but with no significant difference in a metaregression test (*p* = 0.051). There was no significant difference in MSI between EBVaGCs and non-EBVaGCs. CD8+ TILs were significantly higher in EBVaGCs than in non-EBVaGCs. In addition, there was no significant correlation between EBV positivity and loss of E-cadherin (Appendix A).

## 4. Discussion

In other epithelial malignancies, the prevalence of EBV positivity was found to be 26.37%, 33.44%, and 45.37% in breast, cervical, and oral squamous cell carcinomas, respectively [68,69,70]. The range of EBV positivity reported was variable in GC tissues [7,8,9,10,11,12,13,14,15,16,17,18,19,20,21,22,23,24,25,26,27,28,29,30,31,32,33,34,35,36,37,38,39,40,41,42,43,44,45,46,47,48,49,50,51,52,53,54,55,56,57,58,59,60,61,62,63,64,65,66,67]. However, Chen et al. reported that non-neoplastic gastric tissue did not detect EBV positivity [71]. A TCGA study stated that the incidence of EBVaGCs was 9% [2]. Previous meta-analyses have reported the range as 2–20% and 6–33% [72,73]. In addition, the clinicopathological features of EBV positivity in GCs were variable, according to reports [72,73]. Therefore, the impact of variable EBV positivity on the controversy of clinicopathological implications of EBV in GCs needs to be elucidated. The present study includes a detailed meta-analysis of the clinicopathological implications of EBV positivity in GCs.

In the present study, the estimated EBV positive rate was 11.3%. EBV positive rates ranged from 1.2% to 89.2% in the individual eligible studies [7,8,9,10,11,12,13,14,15,16,17,18,19,20,21,22,23,24,25,26,27,28,29,30,31,32,33,34,35,36,37,38,39,40,41,42,43,44,45,46,47,48,49,50,51,52,53,54,55,56,57,58,59,60,61,62,63,64,65,66,67]. In previous meta-analyses, EBV positive rates have been reported as 7.5% and 12.6% in 2010 and 2019, respectively [74,75]. Various factors, including the eligible studies, may have affected the differences of EBV positivity between meta-analyses. In Murphy’s report, a subgroup analysis based on study location was performed, and the estimated EBV positive rates in America, Asia, and Europe were 9.88%, 8.28%, and 8.70%, respectively [72]. In the current study, the positive rate was highest in Asia at 13.8%. However, there were no significant differences between study locations in the metaregression test. Lee et al. reported that locations with a high prevalence of GCs had low EBV positivity [76]. They showed only odds ratios according to study locations, but not the estimated rates. As the criteria of the odds ratio were not described, interpretation of the odds ratio was not possible. They described that the EBV-positive rate of Asians was 8.4% through simple estimation using the raw data of each study. A meta-analysis did not obtain this result. Moreover, the estimated EBV-positive rates of Caucasian and Hispanic patients did not differ from Asians. In another meta-analysis, there was no significant difference in EBV-positive rates between study locations [75].

In addition, EBV positivity rates can differ according to the histologic type of GC. The highest EBV-positive rate was found in GC with lymphoid stroma at 57.3%. The implications of study location and ethnicity on EBV positivity may be less important when compared to the subtype of GC. Furthermore, the impact of studied years can contribute to varying EBV-positive rates. Additionally, we investigated EBV positivity in tubular adenocarcinoma according to study years. Based on 2017 data, EBV-positive rates were 0.113 (95% CI: 0.063–0.195) and 0.375 (95% CI: 0.132–0.703) after 2017 and before 2017, respectively, with a significant difference between subgroups (*p* = 0.012 in a metaregression test; data not shown). The possible causes are different methodologies and different histologic subtypes of the included cases. The cellular component can affect EBV positivity. In GCs, TILs can show EBV positivity [71]. Of course, the use of a PCR method with microdissection is possible for a more detailed examination; however, this limitation cannot be solved by microdissection due to intratumoral and peritumoral lymphocytes. Although PCR methods are more sensitive than in situ hybridization (ISH) methods, EBV positivity should be elucidated by evaluating cellular fractions, such as in ISH [71]. However, a definitive cause for the difference of EBV positivity by study years could not be found.

In previous studies, EBV positivity has been significantly correlated with some clinicopathological characteristics, sex, and tumor location [22,26,53]. In the present study, there was a significant correlation between EBV positivity and the patient’s sex; however, EBV positivity was not correlated with lymphovascular invasion or pTNM stage. The clinicopathological significance of EBV infection is different by reports [24,25,74,75,76]. Huang et al. reported that EBV infection in GCs was correlated with high pTNM stages and lymphatic tumor invasion, as opposed to our results [24,25]. Lee et al. reported that EBV positivity was higher in younger patients than in older patients [76]. Li et al. reported a correlation between EBV positivity and lymph node metastasis [74]. However, other meta-analyses showed no correlation between EBV positivity and lymph node metastasis, in agreement with our result [75,76]. For the evaluation of correlation with lymph node metastasis, Li’s meta-analysis and our meta-analysis included 5 and 40 datasets, respectively. Moreover, they analyzed their data using odds ratios, unlike our analysis. These discrepancies could be involved in the difference of results between the meta-analyses.

Although the molecular characteristics of GCs have been studied [2], previous meta-analyses have not dealt with their correlation with various molecular markers [75]. In our results, CD8+ TILs and PD-L1 expressions of the tumor and immune cells were more frequently found in EBVaGCs than in non-EBVaGCs. Abundant TILs are one of the histologic features in GCs with EBV infection [77,78,79]. In the TCGA report, PD-L1 gene amplification was elevated in EBVaGCs [2]. Furthermore, PD-L1 immunohistochemical expression in tumor cells was more frequently found in EBVaGCs than in non-EBVaGCs [28]. However, the impact of TILs in GCs is not yet fully understood. In addition, further evaluation of the tumor-infiltrating and peritumoral lymphocytes will be needed in GC with lymphoid stroma, which was significantly associated with high EBV positivity. In GCLS, EBV-positive tumors had more PI3K/AKT pathway mutations than EBV-negative tumors [80]. In addition, because EBVaGCs are significantly correlated with high TILs, new immunotherapeutic strategies associated with T-cells are challenging for the treatment of advanced EBVaGCs [81,82]. ARID1A expression was higher in EBVaGCs than in non-EBVaGCs. In the previous meta-analysis, correlations between EBV positivity and molecular markers, such as p53 and CpG island methylator phenotype, were found [76].

This study has some limitations. First, a subgroup analysis based on EBV detection methods could not be performed due to the methods used in the eligible studies. Second, the impact of study years on EBV positivity could not be fully investigated based on subtypes of GCs. We evaluated only tubular adenocarcinomas among the various GC subtypes. Third, the eligible studies used different antibodies and evaluation criteria for immunohistochemistry. However, subgroup analysis based on antibody and evaluation criteria could not be performed due to insufficient information.

## 5. Conclusions

Taken together, our results show that the EBV positivity of GCs is frequently found in male patients and GC with lymphoid stroma. Although EBV positivity was highest in Asians, there was no significant difference between study locations. EBV positivity is significantly correlated with ARID1A and PD-L1 expressions, as well as CD8+ TILs in GCs.

## Figures and Tables

**Figure 1 medicina-56-00345-f001:**
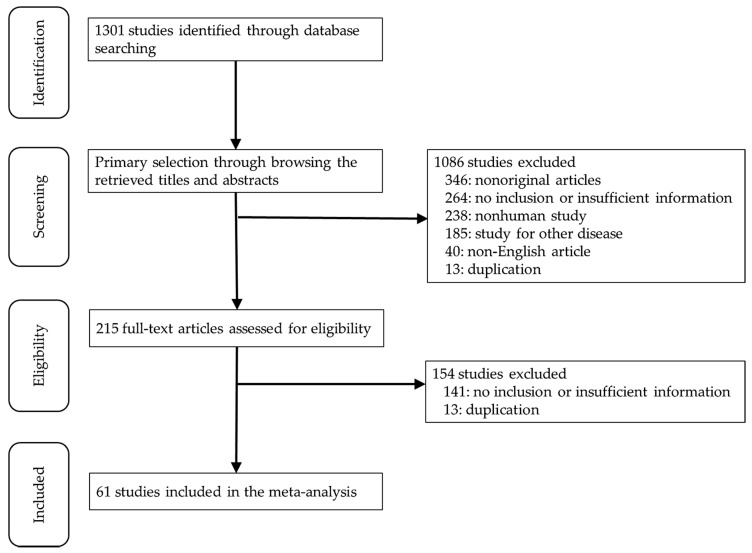
Flow chart of the searching strategy.

**Table 1 medicina-56-00345-t001:** Main characteristics of the eligible studies.

Study	Location	Number of Patients	EBV		Study	Location	Number of Patients	EBV	
			Positive	Negative				Positive	Negative
Ahn 2017	Korea	349	26	323	Ma 2017	China	571	31	540
Castaneda 2019	Peru	375	72	303	Martinez- Ciarpaglini 2019	Spain	209	13	196
Birkman 2018	Finland	238	17	221	Min 2016	Korea	145	124	21
Böger 2017	Germany	484	22	462	Nogueira 2017	Portugal	82	9	73
Bösch 2019	Germany	189	11	178	Noh 2018	Korea	449	36	413
Baek 2018	Korea	276	59	217	Osumi 2019	Japan	898	71	827
Chapel 2000	France	56	7	49	Pereira 2018	Brazil	286	30	256
Cho 2004	Korea	24	19	5	Ramos 2019	Brazil	178	18	160
de Lima 2012	Brazil	160	11	149	Ribeiro 2017	Portugal	179	15	164
De Rosa 2018	Italy	169	33	136	Roh 2019	Korea	582	41	541
de Souza 2014	Brazil	125	12	113	Saito 2017	Japan	232	96	136
de Souza 2018	Brazil	302	62	240	Setia 2019	USA/Korea	486	33	453
Dong 2016	China	855	59	796	Shen 2017	China	202	42	160
Gasenko 2019	Latvia	302	26	276	Shibata 1993	USA	187	19	168
Grogg 2003	USA	110	7	103	Shinozaki 2009	Japan	111	43	68
Guo 2019	China	270	18	252	Sun 2019	China	165	2	163
Han 2016	Korea	410	30	380	Trimeche 2009	Tunisia	96	4	92
Huang 2014	Taiwan	1020	52	968	Truong 2009	USA	235	12	223
Huang 2019	Taiwan	1248	65	1183	Valentini 2019	Italy	70	2	68
Irkkan 2017	Turkey	105	8	97	van Beek 2004	Netherlands	566	41	525
Kawazoe 2017	Japan	487	25	462	Vo 2002	USA	108	11	97
Kawazoe 2019	Japan	225	14	211	Wang 2005	China	58	13	45
Kijima 2003	Japan	420	28	392	Wu 2017	China	340	17	323
Kim 2019 (a)	Korea	273	25	248	Xing 2017	China	967	34	933
Kim 2019 (b)	USA	43	6	37	Yanagi 2019	Japan	1067	69	998
Koriyama 2007	Japan	149	49	100	Zhang 2017	China	218	64	154
Kwon 2017	Korea	394	26	368	Yoon 2019	USA	107	3	104
Leung 1999	China (Hong Kong)	79	18	61	Yen 2014	Brunei Darussalam	81	25	56
Li 2016	China	137	30	107	Zhang 2019	China	1013	58	955
Lim 2017	Korea	241	215	26	Zhou 2019	China	300	28	272
Ma 2016	USA	44	7	37					

EBV, Epstein–Barr virus.

**Table 2 medicina-56-00345-t002:** The estimated rates of Epstein–Barr virus positivity in gastric carcinoma.

	Number of Subsets	Fixed Effect (95% CI)	Heterogeneity Test (*p*-Value)	Random Effect (95% CI)	Egger’s Test (*p*-Value)
EBV positive rate	61	0.116 (0.111, 0.121)	<0.001	0.113 (0.088, 0.143)	0.912
Asia	34	0.121 (0.115, 0.128)	<0.001	0.138 (0.096, 0.194)	0.238
America	13	0.132 (0.118, 0.148)	<0.001	0.103 (0.077, 0.137)	0.002
Europe	12	0.083 (0.073, 0.095)	<0.001	0.080 (0.061, 0.106)	0.558
Africa	1	0.042 (0.016, 0.106)	1.000	0.042 (0.016, 0.106)	-

CI, confidence interval; EBV, Epstein–Barr virus.

**Table 3 medicina-56-00345-t003:** Clinicopathological significance of Epstein–Barr virus positivity in gastric carcinomas.

	Number of Subsets	Fixed Effect (95% CI)	Heterogeneity Test (*p*-Value)	Random Effect (95% CI)	Egger’s Test (*p*-Value)	MRT (*p*-Value)
Age						
EBV-positive	20	61.848 (61.115, 62.581)	<0.001	62.161 (60.126, 64.197)	0.693	0.568
EBV-negative	16	63.532 (63.219, 63.846)	<0.001	63.519 (60.349, 66.690)	0.788	
Male ratio						
EBV-positive	44	0.823 (0.802, 0.843)	0.063	0.824 (0.796, 0.849)	0.189	<0.001
EBV-negative	40	0.638 (0.629, 0.647)	<0.001	0.639 (0.620, 0.658)	0.945	
Size (cm)						
EBV-positive	12	3.840 (3.666, 4.015)	<0.001	4.890 (4.223, 5.556)	<0.001	0.918
EBV-negative	7	4.595 (4.507, 4.683)	<0.001	4.588 (4.354, 4.823)	0.957	
Tumor differentiation, poorly						
EBV-positive	20	0.674 (0.630, 0.716)	0.004	0.682 (0.611, 0.745)	0.514	0.112
EBV-negative	20	0.608 (0.595, 0.622)	<0.001	0.597 (0.525, 0.665)	0.761	
Lymphatic invasion						
EBV-positive	7	0.487 (0.429, 0.546)	<0.001	0.476 (0.299, 0.659)	0.843	0.523
EBV-negative	7	0.498 (0.483, 0.513)	<0.001	0.522 (0.454, 0.588)	0.583	
Vascular invasion						
EBV-positive	7	0.297 (0.249, 0.350)	<0.001	0.286 (0.189, 0.408)	0.636	0.890
EBV-negative	7	0.276 (0.263, 0.290)	<0.001	0.297 (0.202, 0.413)	0.875	
Perineural invasion						
EBV-positive	8	0.415 (0.350, 0.482)	<0.001	0.399 (0.213, 0.619)	0.807	0.094
EBV-negative	8	0.517 (0.498, 0.535)	<0.001	0.521 (0.458, 0.584)	0.875	
Low pT stage (pT1/T2)						
EBV-positive	33	0.435 (0.401, 0.471)	<0.001	0.366 (0.274, 0.469)	0.066	0.670
EBV-negative	31	0.413 (0.402, 0.424)	<0.001	0.350 (0.283, 0.422)	0.141	
Lymph node metastasis						
EBV-positive	40	0.493 (0.461, 0.526)	<0.001	0.595 (0.496, 0.686)	0.014	0.127
EBV-negative	37	0.593 (0.583, 0.604)	<0.001	0.655 (0.595, 0.711)	0.064	
pTNM stage						
EBV-positive	25	0.507 (0.469, 0.544)	<0.001	0.500 (0.419, 0.580)	0.738	0.236
EBV-negative	25	0.451 (0.439, 0.463)	<0.001	0.460 (0.425, 0.496)	0.411	

CI, confidence interval; MRT, metaregression test; EBV, Epstein–Barr virus.

**Table 4 medicina-56-00345-t004:** The estimated rates of Epstein–Barr virus positivity in gastric carcinomas according to the histologic types.

Histologic Type	Number of Subsets	Fixed Effect (95% CI)	Heterogeneity Test (*p*-Value)	Random Effect (95% CI)	Egger’s Test (*p*-Value)
Tubular adenocarcinoma	6	0.152 (0.132, 0.174)	<0.001	0.174 (0.086, 0.320)	0.531
Poorly cohesive carcinoma	8	0.102 (0.063, 0.160)	0.038	0.078 (0.033, 0.173)	0.263
Mixed carcinoma	4	0.043 (0.016, 0.109)	0.306	0.039 (0.013, 0.113)	0.054
Papillary carcinoma	2	0.022 (0.004, 0.101)	0.530	0.022 (0.004, 0.101)	-
Mucinous carcinoma	4	0.053 (0.013, 0.190)	0.688	0.053 (0.013, 0.190)	0.042
GCLS	5	0.576 (0.468, 0.676)	0.203	0.573 (0.428, 0.706)	0.748
Solid carcinoma	2	0.130 (0.046, 0.316)	0.828	0.130 (0.046, 0.316)	-
Undifferentiated carcinoma	1	0.111 (0.015, 0.500)	1.000	0.111 (0.015, 0.500)	-

CI, confidence interval; GCLS, gastric carcinoma with lymphoid stroma.

**Table 5 medicina-56-00345-t005:** The estimated rates of various markers in gastric carcinomas according to the Epstein–Barr virus positivity.

Markers	Number of Subsets	Fixed Effect (95% CI)	Heterogeneity Test (*p*-Value)	Random Effect (95% CI)	Egger’s Test (*p*-Value)	MRT (*p*-Value)
PD-L1 in tumor cells						
EBV-positive	14	0.500 (0.447, 0.554)	<0.001	0.573 (0.449, 0.688)	0.047	<0.001
EBV-negative	14	0.337 (0.323, 0.352)	<0.001	0.183 (0.118, 0.272)	0.008	
PD-L1 in immune cells						
EBV-positive	8	0.610 (0.531, 0.683)	<0.001	0.832 (0.630, 0.935)	0.007	0.002
EBV-negative	8	0.572 (0.552, 0.592)	<0.001	0.487 (0.357, 0.619)	0.081	
p53 overexpression						
EBV-positive	5	0.359 (0.256, 0.477)	0.223	0.194 (0.067, 0.446)	0.023	0.090
EBV-negative	4	0.464 (0.418, 0.511)	<0.001	0.439 (0.314, 0.572)	0.502	
ARID1A						
EBV-positive	4	0.295 (0.206, 0.403)	0.309	0.295 (0.196, 0.418)	0.519	0.021
EBV-negative	4	0.176 (0.153, 0.201)	0.055	0.170 (0.134, 0.214)	0.530	
HER2						
EBV-positive	8	0.048 (0.024, 0.093)	0.723	0.048 (0.024, 0.093)	0.167	0.051
EBV-negative	8	0.101 (0.088, 0.115)	<0.001	0.104 (0.070, 0.152)	0.739	
Microsatellite instability						
EBV-positive	5	0.087 (0.040, 0.179)	0.240	0.077 (0.028, 0.190)	0.230	0.536
EBV-negative	5	0.104 (0.089, 0.121)	<0.001	0.108 (0.069, 0.166)	0.637	
CD8+ TILs						
EBV-positive	4	0.705 (0.584, 0.802)	0.100	0.761 (0.547, 0.894)	0.163	0.001
EBV-negative	4	0.307 (0.275, 0.341)	<0.001	0.269 (0.141, 0.450)	0.851	

CI, confidence interval; MRT, metaregression test; PD-L1, programmed death-ligand 1; EBV, Epstein–Barr virus; ARID1A, AT-rich interactive domain-containing protein 1A; HER2, human epidermal growth factor receptor 2; TIL, tumor-infiltrating lymphocyte.

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
