# Peer review of "Clinicopathological Significance of EBV-Infected Gastric Carcinomas: A Meta-Analysis"

_medicina, 2020, doi:10.3390/medicina56070345_

Round 1

Reviewer 1 Report

The authors examined clinicopathological significances of EBV-infected Gastric Carcinomas by a meta-analysis. Their results are almost identical to those described in previous reports. The results newly identified by them are part of molecular markers. Their results are interesting, but there are some problems with the method.

  1. Their data on some molecular markers in EBVaGC is the most important in this study. I know that PD-L1 expression rate in EBVaGC is various in each study. Are their criteria of positive expression consistent? I would like to ask to indicate their criteria of positive expression in each gene. The method for immunohistochemistry is also important. At least, antibodies they used should be described. I think it would be difficult to compare data examined by different criteria and method.
  2. Pathological findings are also very important. I have the same concerns as findings of molecular markers. Are the results of tumor-infiltrating lymphocytes (TILs) and GCLS evaluated based on the same diagnostic criteria?

Author Response

We tried to address the points raised by the reviewers as best as we can. The specific responses to the reviewers’ comments are described in Reply to Reviewers. We also fixed other unintended errors in our manuscript.

Reviewer 2 Report

Comments for the authors:-

EBV–associated gastric carcinoma (EBVaGC) which has not received much clinical attention, are common among EBV-associated malignancies and their rates are currently increasing each year and recuires immediate attention for better diagnosis and therapeutic outcome of these patients in clinic. As such, understanding the clinical and pathological features of EBVaGC is critical to determine the mode of therapy in these patients. In this manuscript, Pyo et al. have demonstrated the clinicopathologic 17 significances of Epstein-Barr virus (EBV) infection in gastric carcinomas (GCs) through a meta-analysis. Overall the mansucript is well composed. However, according to this reviewer, the authors must considers to address the following queries to make the manuscript more feasible to broader audience.

Comments:

  1. The authors should describe the incidence rate of EBV-associated malignancies.
  2. The authors should also discuss the current immuno-therapeutic options available including T cell therapy and prophylactic vaccines. They must cite PMID: 30987475
  3. The authors must consider keeping their statistics consistent. The authors highlight in Line 21: “The estimated 22 EBV-infected rate of GC was 0.113 (95% confidence interval (CI) 0.088-0.143). The EBV infected rates 23 in GC were 0.138 (95% CI 0.096-0.194), 0.103 (95% CI 0.077-0.137), 0.080 (95% CI 0.061-0.106), and 24 0.042 (95% CI 0.016-0.106) in Asia, America, Europe, and Africa” while in Line 97, they against state that: “First, we investigated and analyzed the EBV positive rates of GCs. The estimated EBV positive 98 rate was 11.3% (95% CI: 8.8%–14.3%) in overall GC cases (Table 2). In the subgroup analysis based on 99 study location, the EBV infected rate was the highest in Asia, compared to that in other regions. The 100 EBV infected rate in the Asia region was 13.8% (95% CI: 9.6%–19.4%)”. This variation in statistics will confuse readers and must be kept consistent throughout the manuscript.
  4. The manuscript should be proof read by a native English speaker prior to re-submission.

Author Response

(The authors gave the same response as above.)

Reviewer 3 Report

The topic of the paper is intriguing and somewhat important for daily clinical practice.

Authors carried out a meta-analysis on a total of 61 studies to investigate the role of EBV infection within patients affected by gastric cancer.

The study is interesting and properly referenced, with a clear aim, satisfactory statistical analyses– albeit my knowledge is far from statistics. However, I find the “introduction” section too short and it should be expanded (suggestions reported below); “material and method” and “results” sections are exhaustive although not clear in several parts. Lastly, I found the discussion session a bit simple (suggestions here below) and a mere report of results. Authors neither do discuss potential implications of your results nor the benefits of study in the clinical research (for instance: you do not discuss results on ARID1A: what is the possible implication of its higher expression in patients with EBVaGC?).

The overall English style is really poor and requires important editing revision. The writing style is choppy without flow and does not hold complex thought.

The paper needs extensive major revisions before being suitable for publication.

I have reported suggested changes here below, divided into major and minor points; after these changes, I would be happy to review the work.

Major points: 

Lines 42-43: To expand the introduction, I strongly suggest to describes currently available therapies and histological subtypes of gastric cancer, maybe using a table. In doing so, please use these 2 papers recently published:

  1. Rodriquenz MG, Roviello G, D'Angelo A, Lavacchi D, Roviello F, Polom K. MSI and EBV Positive Gastric Cancer's Subgroups and Their Link With Novel Immunotherapy. J Clin Med. 2020;9(5):1427. Published 2020 May 11. doi:10.3390/jcm9051427
  2. Fang WL, Chen MH, Huang KH, et al. The Clinicopathological Features and Genetic Alterations in Epstein-Barr Virus-Associated Gastric Cancer Patients after Curative Surgery. Cancers (Basel). 2020;12(6):E1517. Published 2020 Jun 10. doi:10.3390/cancers12061517

Lines 58-59: “The titles and abstracts of all searched articles were screened for 58 inclusion and exclusion.” What do you mean with inclusion and exclusion? Not clear

Table 2:

  1. There is no correspondence between figures reported within the text (lines 98-101) and table – please use the same unit of measurement. It’s confusing
  2. Why did you divide studies based on location?

Table 3:

  1. As in Table 2, there is no correspondence between figures in text and table (82% à82); it’s confusing
  2. Line 105 = “Other clinicopathological characteristics” – please indicate
  3. What do you mean for “Number of subsets”? It is not clear
  4. The last column reports MRT value: what is it? You did not mention it in your “materials and methods” section

Table 5:

  1. What did you investigate PD-L1 in immune cells rather than PD-1?
  2. Why did you investigate AID1A? I think it’s better to briefly introduce this gene/protein and explain why it’s important for you. This gene is not very famous yet.
  3. Instead of “Micro”, I would suggesting reporting “MSI” as a group name.

Lines 128-129: “ non-neoplastic gastric tissue did not detect EBV positivity [68]”- Who does this sentence refer to? Your study or the referenced one? It is not clear.

Lines 155-157: Here you say: “after 2017 and before 2017” – I don’t see these data in the “results section”

Line 161: “due to TILs”. This sentence is truncated. Due to TILs what?

Line 168: What do you mean with “tumour behaviours”?

Line 169: “significance of EBV infection is controversial” – what do you mean?

Minor points:

Line 24:  “ The EBV infected rates 22 in GC were” . Please improve the English style. Not very clear.

Lines 36-39: To help the reader, please try to bind at least to sentences using linkers. Reding doesn’t flow.

Lines 49-53: Please improve English style

Line 90: Please remove the comma after (n=40)

Line 95: Table 2 not labelled

Line 103: Please improve English style

Line 134: I would use “included” rather than “comprised”

Line 139: Please modify this repetition “included including”

Lines 143 and 144: Both sentences start with “However”. Please modify.

Line 170: Please improve English style

Supplementary materials are not cited in manuscript

Author Response

(The authors gave the same response as above.)

Round 2

Reviewer 1 Report

I think that it is difficult to combine data evaluated by different methods and criteria by meta-analysis.

Reviewer 3 Report

Well done!